# Uncommon Presentation of Pityriasis Rubra Pilaris of the Scalp: Clinical, Trichoscopic, and Histopathologic Features and Review of the Literature

**DOI:** 10.3390/medicina60111839

**Published:** 2024-11-08

**Authors:** Michela Starace, Stephano Cedirian, Federico Quadrelli, Tullio Brunetti, Lidia Sacchelli, Cosimo Misciali, Giacomo Clarizio, Pietro Sollena, Francesco Tassone, Iria Neri, Bianca Maria Piraccini

**Affiliations:** 1Dermatology Unit, IRCCS Azienda Ospedaliero-Universitaria di Bologna, Policlinico S. Orsola-Malpighi, 40138 Bologna, Italy; michela.starace2@unibo.it (M.S.); federico.quadrelli@studio.unibo.it (F.Q.); tullio.brunetti@studio.unibo.it (T.B.); lidia.sacchelli@gmail.com (L.S.); cosimo.misciali@unibo.it (C.M.); giacomo.clarizio@studio.unibo.it (G.C.); iria.neri@aosp.bo.it (I.N.); biancamaria.piraccini@unibo.it (B.M.P.); 2Department of Medical and Surgical Sciences, Alma Mater Studiorum University of Bologna, 40138 Bologna, Italy; 3Dermatology Unit, Dipartimento di Scienze Mediche e Chirurgiche Addominali ed Endocrino Metaboliche, Fondazione Policlinico Universitario A, Gemelli—IRCCS, 00168 Rome, Italy; pietrosollena@gmail.com (P.S.); france.tassone@libero.it (F.T.)

**Keywords:** pityriasis rubra pilaris, trichoscopy, histopathology, hair disease

## Abstract

Pityriasis rubra pilaris (PRP) presents a diagnostic challenge due to its varied clinical manifestations and the scarce literature on scalp involvement. This article presents a case report of a 59-year-old female with PRP solely affecting the scalp, detailing its clinical, trichoscopic, and histopathological features. Trichoscopy revealed a novel finding of white-silvery scales forming hair casts with a triangular shape, distinct from the existing literature. A literature review comparing our findings with pertinent articles underscored the uniqueness of our case. We discuss differential diagnoses and treatment options, based on available evidence. Our case highlights the importance of understanding scalp manifestations in PRP, enhancing diagnostic accuracy, and improving treatment strategies for this rare condition. Furthermore, the review of the literature compares our observations with available case reports and case series, outlining differential diagnoses and trichoscopic and histopathological diagnostic approaches to PRP, enriching overall clinical knowledge of PRP.

## 1. Introduction

Pityriasis rubra pilaris (PRP) is an idiopathic inflammatory skin condition first described by Claudius Tarral in 1835, who recognized it as a variant of psoriasis [1,2]. Later, in 1856, Alphonse Devergie coined the term “pityriasis pilaris” to describe this condition [1,2].

PRP is an inflammatory skin condition with uncertain pathogenetic pathways. Autoinflammatory diseases (AIDs) and autoimmune diseases (ADs) are distinct yet overlapping immune-mediated disorders with different molecular bases [3,4,5]. ADs generally stem from a loss of immune tolerance, activating adaptive immune responses against self-antigens, while AIDs are mainly driven by dysregulation of the innate immune system [3,4,5]. AIDs often involve inflammasomes, such as the NLRP3 complex, which activates inflammatory cytokines IL-1β and IL-18 [3,4]. Inflammasomes are large intracellular protein complexes that convert inactive IL-1β into its inflammatory form. Excessive IL-1β from overactive inflammasomes leads to inflammation and fever [3,4]. At least seven inflammasome complexes exist, each centered around a unique danger-sensing protein that triggers assembly upon activation. Mutations in these complexes can cause inflammasomopathies (e.g., familial Mediterranean fever (FMF) and hyperimmunoglobulin D (IgD) syndrome) [3,4]. These diseases commonly manifest with fever, rash, joint pain, and chest or abdominal discomfort [3,4]. Some AIDs also involve interferons, ubiquitination, or immunodeficiencies [3,4,6,7]. By contrast, ADs are closely linked to adaptive immune responses and cytokine pathways like IL-17 and IL-23, also implicated in psoriasis [5]. Pityriasis rubra pilaris (PRP), sometimes classified as a psoriasis variant, may involve the IL-23/Th17 axis, as suggested by its response to IL-17 and IL-23 therapies [8,9]. In a study of 11 PRP patients, those responding well to IL-17A inhibition showed normalized expression of CCL20, IL17C, and IL23A, while poor responders had persistently high levels of these markers [8]. On the other hand, innate pathway activation has been observed in poor responders [9]. Despite some overlap, PRP is more closely associated with ADs than with AIDs.

*CARD14* has been identified as a pathogenic gene variant in families with autosomal dominant PRP [10,11,12,13]. Located within the *PSORS2* gene locus, *CARD14* is also associated with familial pustular psoriasis [11]. Studies using two mouse models that mimic human *CARD14* variants have provided insight into the disease mechanism, as these models displayed skin scaling and erythema [12,13]. The genetic variations cause constant activation of the *CARD14* gene product, CARMA2, leading to elevated nuclear factor kappa B (NF-κB) activity and an increase in the chemokine CCL20, a strong activator of the Th17 pathway [12,13]. Treatment with an anti-IL-23p19 antibody showed an improvement in skin condition [13].

Finally, as previously mentioned, similar pathogenic mechanisms may link PRP and psoriasis [11,14]; in one study, all dysregulated genes in PRP skin were also present in psoriasis skin, with no genes unique to PRP [14].

PRP is characterized by its diverse clinical presentations and can affect individuals across various ethnicities and genders [2,15]. The global incidence of PRP varies significantly. In Great Britain, the condition is estimated to occur in approximately 1 in 5000 new cases, whereas in India, the incidence is lower, around 1 in 50,000 [1,16]. Among children, PRP is somewhat more common, with a frequency of about 1 in 500. The condition exhibits a bimodal age distribution, meaning that it primarily affects two distinct age groups: individuals in the first decade of life and those in their fifth to sixth decades [1,2,15].

The pathophysiology of PRP remains enigmatic, which contributes significantly to the uncertainty surrounding its causes or triggers [1,16]. Despite extensive research, a definitive mechanism has yet to be identified, leaving the door open for various hypotheses. In a comprehensive review conducted by Zhou et al., several potential triggers for PRP were identified, with particular emphasis on COVID-19 vaccination, malignancy, and COVID-19 infection [17]. These findings suggest that clinicians should maintain a high index of suspicion for these factors when diagnosing and treating PRP or similar skin eruptions. COVID-19 and its associated vaccinations are particularly noteworthy, given their widespread impact and the temporal correlation observed in some PRP cases. The possibility that the immune response triggered by COVID-19 infection or vaccination might play a role in the onset of PRP adds a layer of complexity to its management and highlights the need for further research in this area. Another significant consideration is the potential link between PRP and underlying malignancies. The review by Zhou et al. positions malignancy as the second most commonly reported trigger of PRP, accounting for approximately 8.2% to 10% of cases in three separate case series [17]. This association raises the possibility that PRP could be a paraneoplastic syndrome, where the skin eruption serves as a cutaneous marker of an underlying cancer. Although recent studies have not provided extensive details on the specific types of malignancies associated with PRP, the review identified nine different cancers, including chronic lymphocytic leukemia, cholangiocarcinoma, and prostate cancer, among others [17]. The potential association between PRP and malignancy underscores the importance of thorough evaluation and vigilance on the part of clinicians. When encountering PRP or PRP-like eruptions, it may be prudent to consider an underlying malignancy, particularly in cases that are atypical or refractory to standard treatments [17]. The possibility of PRP serving as a marker for undiagnosed cancer highlights the need for a multidisciplinary approach to patient care, integrating dermatologic assessment with oncologic evaluation to ensure comprehensive management. The review by Zhou et al. clarifies the need for further research to better understand the link between PRP and malignancy and to identify potential mechanisms that could explain this association [17].

Given the observed similarities between pityriasis rubra pilaris (PRP) and psoriasis, the potential connection between PRP and malignancy may arise from similar underlying mechanisms to those seen in psoriasis [11,14]. Both conditions share a chronic inflammatory environment that, in PRP, is characterized by elevated levels of Th17 and Th1 cytokines in both the systemic circulation and lesional skin. Key cytokines involved include IL-17A, IL-17F, IL-22, TNF, IL-6, IL-12, IL-23, and IL-1β, which could contribute to the development of malignancy [18,19]. Additionally, regulatory T cells, particularly Th2 cells, can be activated by inflammation and may secrete pro-tumorigenic factors such as transforming growth factor (TGF), IL-4, IL-6, and IL-10, thus perpetuating chronic inflammation [18]. T cells can attract and activate B cells, which, in turn, promote carcinogenesis while simultaneously depleting cytotoxic T lymphocytes (CTLs) [19]. Given that the skin is continually exposed to various environmental stimuli, this inflammatory response can initiate multiple processes that affect the cellular components of the skin and contribute to carcinogenesis [18]. Moreover, besides the above-mentioned persistent low-grade inflammation and the elevated levels of circulating cytokines, the hyperproliferative nature of keratinocytes in PRP could also increase the risk of malignancy in these patients [18,19].

To better understand and manage the diverse manifestations of PRP, Griffiths introduced a classification system in 1980 that categorizes the condition into five distinct subtypes [2]. This classification is based on factors such as age of onset, clinical morphology, disease course, and prognosis [2]. The five subtypes are as follows: classic adult type, atypical adult type, classic juvenile type, circumscribed juvenile type, atypical juvenile type. To these, a sixth subtype specific to HIV-infected individuals has been added [1,2,16]. The classical adult subtype (type I) is the most common (more than 50% of reported cases), characterized by sudden onset primarily affecting the scalp and upper body, extending to the trunk and limbs, featuring red-orange keratotic follicular papules merging into plaques. Nail changes and ectropion are common, with around 80% experiencing spontaneous remission within three years [1]. The atypical subtype (type II) is less common, and it is usually characterized by ichthyosiform lesions, with areas of eczematous change and possible scarring alopecia. Type II is difficult to treat and often tends to become a chronic intractable disease. The classical juvenile subtype (type III) shows clinical and prognostic analogous characteristics to type I PRP, but its onset is within the first 2 years of life [1]. The most frequent PRP in young patients is the circumscribed juvenile subtype (type IV) occurring in prepuberal children and characterized by clearly demarcated areas of follicular hyperkeratosis and erythema mainly located on the knees and the elbows [1]. The atypical juvenile subtype (type V) is a less common generalized type of juvenile PRP, with an early onset and a chronic course. It is characterized by prominent follicular hyperkeratosis, scleroderma-like changes on the palms and the soles, and infrequent erythema. Type VI is similar to type I but affects HIV-positive patients [1,2,15,16].

Our clinical approach for a patient with suspected PRP typically includes a thorough medical history, detailed clinical examination to identify any cutaneous manifestations, trichoscopy for cases with scalp involvement, and dermoscopy-guided biopsy to exclude other mimicking conditions [1,2]. Generally, aside from basic blood tests (e.g., complete blood count, ESR, and CRP), we do not conduct additional screening tests, as associations with malignancies and other conditions remain inconclusive. Instead, we reserve more extensive screening for patients who present with signs and symptoms unrelated to PRP, such as weight loss, fever, or arthralgia [15,16,20]. However, some authors recommend HIV screening for adolescents and young adults due to the potential for PRP type VI. Additionally, for cases with onset in the first year of life, or in children with refractory PRP or other indicators suggestive of CARD14-associated papulosquamous eruption (CAPE), genetic testing for CARD14 gene variations is advisable [20].

Although there is documented evidence of scalp involvement in PRP, particularly when it is associated with involvement of the trunk, the literature on its specific clinical manifestations, trichoscopic features, and histopathological characteristics remains relatively sparse [21,22,23,24,25,26]. In this report, we present a unique case of PRP that exclusively affects the scalp. We provide a detailed description of the clinical presentation, trichoscopic findings, and histopathological features associated with this condition. Additionally, we explore the differential diagnoses to consider, outline potential treatment options, and perform a comprehensive literature review. This review aims to juxtapose our observations with existing data, thereby contributing to a more nuanced understanding of scalp PRP and enhancing clinical knowledge in this area.

## 2. Case Report

We present the case of a 59-year-old female patient who sought consultation at the Dermatology Unit of Policlinico S. Orsola-Malpighi in Bologna, due to the sudden onset of widespread scaling on her scalp. Despite the extensive nature of the scalp lesions, the patient was entirely asymptomatic, reporting no discomfort or itching. The clinical symptoms had developed rapidly, within just two weeks before her visit. The patient had no prior history of similar scalp issues and reported no family history of similar conditions. Additionally, she denied having any other dermatological or systemic diseases and was not on any medications at the time of her visit.

Upon examination, trichoscopy revealed notable findings, including closely adherent white-silvery scales that formed hair casts with a distinctive triangular shape, resembling an “arrowhead” pointing towards the scalp. This particular trichoscopic feature was striking and had not been previously documented in the literature. In addition to these unique scales, trichoscopy also identified a few dilated and circular blood vessels, adding to the diagnostic complexity [Figure 1A,B].

To further substantiate the diagnosis, a histopathological analysis was performed. The histologic examination of the scalp biopsy revealed key features consistent with PRP. Specifically, the sample showed dilated infundibula, which were plugged by alternating orthokeratotic and parakeratotic cells. Additionally, there was evidence of superficial perivascular dermatitis, a finding that supports the inflammatory nature of the condition [Figure 2A,B]. Taken together, the clinical presentation, trichoscopic findings, and histological features led to a conclusive diagnosis of scalp PRP. We treated the patient with daily occlusive clobetasol cream for two months to dissolve the scales and reduce inflammation. At the two-month follow-up, we re-evaluated the patient, and, as inflammation had not significantly decreased, we initiated treatment with acitretin. By the four-month follow-up, the clinical picture had improved significantly, showing a notable reduction in diffuse desquamation.

This case highlights the importance of a thorough and multidisciplinary approach to diagnosing rare dermatological conditions, particularly when they present with unusual or unique features.

## 3. Discussion

PRP is a chronic inflammatory skin disorder characterized by a wide array of clinical manifestations, which can vary significantly among individuals [27,28]. Despite extensive research, the precise etiology and pathogenesis of PRP remain largely unknown, making it a complex condition to diagnose and treat [27,28,29]. The onset of PRP can be associated with various triggering events, including infections, malignancies, and autoimmune diseases, suggesting that external or internal factors may play a role in its development [20,27]. Additionally, in some cases, a hereditary component has been observed, with mutations in the *CARD14* gene being linked to familial forms of PRP [10,11,12,13,30]. However, it is important to note that PRP does not appear to have a preference for any particular gender or age group, as cases have been reported across all demographics [27,28,29,30,31].

We undertook a comprehensive literature review to investigate the relationship between scalp involvement in PRP and the application of trichoscopy and histopathology in evaluating this condition. To achieve this, we systematically searched the PubMed, EMBASE, and MEDLINE databases, covering literature published from 1957 through January 2024. Our search strategy included specific keywords, namely “pityriasis rubra pilaris AND scalp”, “pityriasis rubra pilaris AND alopecia,” and “pityriasis rubra pilaris AND trichoscopy”, to ensure a focused exploration of relevant studies. We limited our search to articles published in English to maintain consistency in the data reviewed and to ensure the relevance of the findings to an English-speaking audience. Additionally, we took care to eliminate any duplicate titles that appeared across the different databases to avoid redundancy in our analysis. Through this process, we identified a total of six articles that were pertinent to our review. These included four case reports and two case series, which provided valuable insights into the clinical features, scalp involvement, and diagnostic techniques related to PRP [21,22,23,24,25,26].

While the studies we reviewed did examine scalp involvement in cases of PRP, they exhibited notable differences when compared to the unique aspects of our case report. Specifically, two of the articles focused on cases of scarring alopecia associated with PRP, highlighting a more severe and potentially irreversible type of hair loss [23,25]. Another two articles dealt with scalp involvement in the erythrodermic variant of PRP, which is characterized by widespread erythema and scaling of the skin, including the scalp [24,26]. One study reported on a hyperkeratotic manifestation of PRP affecting the scalp in a patient with HIV, illustrating how underlying conditions might influence the presentation of PRP [22]. The final article centered on significant pityriasiform scalp scaling, a condition where the scalp develops thick, flaky skin that can resemble dandruff or psoriasis, complicating the clinical picture [21]. Of particular interest, only the studies by Golinska et al. and Sławinska et al. provided detailed descriptions of the trichoscopic features observed in their cases, offering insights into the use of trichoscopy as a diagnostic tool in PRP [19,24]. Meanwhile, three of the reviewed articles included descriptions of histopathological features, contributing to a better understanding of the microscopic changes in the scalp tissue associated with PRP [22,23,25]. The key findings from these studies are comprehensively summarized in Table 1.

In contrast to the cases documented in the literature, our case of scalp PRP presents a distinct clinical manifestation characterized by diffuse scaling. This presentation is notably devoid of the more severe consequences potentially associated with scalp PRP, such as scarring alopecia, hyperkeratotic plaques, or erythrodermic involvement. These features differentiate our case from previously reported cases, where such complications were more prominent. Trichoscopic examination of our patient revealed features that partially align with those described by Golinska et al. and Sławinska et al., particularly in terms of the general patterns observed in PRP [21,26]. However, our case also presents a unique trichoscopic feature that, to the best of our knowledge, has not been previously reported. Specifically, we observed closely adherent white-silvery scales forming hair casts that exhibit a distinctive triangular shape, reminiscent of an “arrowhead” pointing towards the scalp. This unique trichoscopic appearance represents a novel finding in the assessment of PRP, adding a new dimension to the understanding of its clinical manifestations. Additionally, our histological findings show patterns consistent with those reported in the literature, although it is important to highlight that the scarring variants of PRP typically exhibit stromal fibrosis—a feature that was notably absent in our case. This further underscores the distinctiveness of our case, which, while sharing some common histopathological features with previously reported cases, also diverges in significant ways, as summarized in Table 1.

In evaluating our case, we considered some differential diagnoses, including scalp psoriasis (SPso), seborrheic dermatitis (SD), and pseudotinea amiantacea (PA). Each of these conditions presents with distinct clinical, trichoscopic, and histopathological features, which are summarized in Table 2.

Scalp psoriasis (SPso) is typically characterized by the presence of thick, white-silvery scales that are regularly distributed across the scalp. Trichoscopically, SPso is often identified by the presence of regularly arranged dotted vessels, commonly referred to as “red dots”, within the affected areas. These red dots are a hallmark of SPso, indicating the presence of dilated capillaries in the dermal papillae. In addition to the dotted vessels, SPso may also exhibit globular and looped vessels, and in some cases, there may be a loss of follicular openings due to the thick scaling [32,33,34].

Seborrheic dermatitis (SD), on the other hand, presents with fine, greasy, yellowish scales that are more loosely adherent compared to the thick scales seen in psoriasis. Trichoscopic examination of SD reveals several distinguishing features, including the presence of follicular plugs and yellowish crusts. Another characteristic feature of SD is the presence of dotted and linear vessels, which help to differentiate it from other scalp disorders [34,35].

Pseudotinea amiantacea (PA) is identified by its characteristic tightly adherent, thin, silver-white scales that encase the hair shafts, forming a thick covering that often makes the hair appear matted or tangled. The scales in PA are usually more adherent to the hair shafts than in other conditions, making them a key diagnostic feature. Unlike psoriasis and seborrheic dermatitis, PA does not typically present with significant vascular changes on trichoscopy, but the presence of these tightly adherent scales is distinctive enough to set it apart from other scalp conditions [36,37].

By carefully analyzing these trichoscopic features, we were able to distinguish our case of PRP from these differential diagnoses, each of which presents with a unique set of signs that can be identified through trichoscopy [32,33,34,35,36,37].

The diagnosis of scalp PRP necessitates a thorough evaluation that integrates clinical examination, trichoscopic analysis, and histopathological investigation. Clinically, PRP can present with a range of scalp manifestations, making it essential to differentiate it from other similar conditions through detailed observation and diagnostic tools [1,2,15,16,17,20,21,22,23,24,25,26,27,28,29,30,31]. Trichoscopy plays a vital role in this process by revealing specific features unique to PRP, such as the distinctive scales and vascular patterns that may not be evident in other scalp disorders. Histopathological examination further corroborates the diagnosis by identifying characteristic microscopic changes in the skin [1,2,15,16,17,20,21,22,23,24,25,26,27,28,29,30,31].

In terms of treatment, managing PRP often involves a combination of topical and systemic therapies. Topical treatments, including emollients, corticosteroids, and retinoids, are commonly employed to alleviate symptoms and reduce inflammation. These treatments are generally considered first-line options due to their accessibility and ease of application. However, the response to topical therapies can vary, and in some cases, may be insufficient to control the disease [38,39,40,41,42]. Systemic treatments are often considered when topical therapies do not provide adequate relief. Retinoids, methotrexate, and ultraviolet (UV) therapy are among the systemic options that have been used to manage PRP. Despite their widespread use, the evidence supporting the efficacy of these treatments remains limited, with few large-scale studies available to establish definitive guidelines. This underscores the need for further research to better understand the optimal management strategies for PRP [38,39,40,41,42]. In recent years, biologic agents have emerged as a promising option for treating PRP, particularly in cases resistant to conventional therapies. These agents, which include tumor necrosis factor (TNF) inhibitors, interleukin (IL)-12/IL-23 inhibitors, and IL-17 inhibitors, work by targeting specific pathways involved in the pathogenesis of PRP. Case reports and small case series have demonstrated varying degrees of success with biologics such as infliximab, etanercept, adalimumab, ustekinumab, and secukinumab. These agents have shown potential in reducing disease severity and improving patient outcomes by directly modulating the immune response that drives PRP [38,39,40,41,42]. Ronnenberg et al. proposed a treatment algorithm to guide the management of PRP, advocating for a stepwise approach. According to this algorithm, the treatment should begin with topical therapies as the first-line intervention. If the response is inadequate, systemic retinoids should be considered. Biologic agents are recommended as a subsequent step if conventional treatments fail to achieve the desired results. This algorithm also emphasizes the importance of regular reassessment of the patient’s response to treatment, allowing for timely adjustments to the therapeutic plan based on the evolving clinical picture [38]. Janus kinase (JAK) inhibitors have emerged as a promising treatment option for pityriasis rubra pilaris (PRP), especially in cases where traditional therapies have proven insufficient. The use of JAK inhibitors in PRP is highlighted by several clinical case reports that underscore their potential efficacy in managing complex dermatological conditions [41,42]. One such case was presented by Xiaoyuan et al., involving a 13-year-old girl who had been diagnosed with generalized pustular psoriasis (GPP) and PRP. Her condition was resistant to conventional treatments and the biologic agent secukinumab, showing only partial improvement. However, upon switching to the JAK inhibitor upadacitinib, a remarkable clearance of lesions was observed within just one month. This rapid and significant response suggests that upadacitinib may be an effective treatment for PRP, particularly in patients with overlapping or complex dermatological diagnoses where other therapies have failed [41]. Similarly, Tan et al. reported on a 39-year-old female patient with PRP who did not respond adequately to traditional treatments. The patient experienced significant improvement after being treated with tofacitinib, another JAK inhibitor. The effectiveness of tofacitinib is thought to be linked to its ability to inhibit the JAK/STAT pathway, which plays a crucial role in the inflammatory processes underlying PRP. This pathway’s inhibition appears to reduce the inflammatory response, leading to clinical improvement in patients with PRP [42]. These case reports collectively suggest that JAK inhibitors, such as upadacitinib and tofacitinib, could represent a new and promising therapeutic avenue for PRP, particularly in cases resistant to standard therapies. However, while these individual cases are encouraging, the need for larger, more robust clinical studies is essential to confirm the efficacy and safety of JAK inhibitors in a broader PRP patient population. Such studies would help to establish these agents as a standard treatment option and provide clearer guidelines on their use in clinical practice.

Future research on PRP, particularly cases with unusual scalp involvement, could focus on understanding the molecular and genetic underpinnings of this rare presentation. Studies examining genetic mutations, such as those affecting the *CARD14* gene, may help uncover predispositions to scalp-specific PRP and provide insights into its pathogenesis. Expanding research on cytokine and immune signaling, specifically the roles of IL-17, IL-23, and other inflammatory markers, could reveal pathways unique to PRP and guide more targeted therapies. Additionally, trichoscopy, as a diagnostic tool, warrants further exploration to identify specific trichoscopic features of PRP and its differential points from other scalp conditions, like SPso and SD. Longitudinal studies documenting patient responses to both traditional and emerging treatments, including biologics and JAK inhibitors, could provide critical data for developing standardized treatment algorithms. Lastly, examining the correlation between PRP and potential triggering factors, such as autoimmunity, malignancy, and COVID-19, could enhance our understanding of external or internal influences on disease progression and improve preventative care approaches for PRP patients.

## 4. Conclusions

In conclusion, this case report underscores the necessity for developing valid diagnostic and therapeutic algorithms grounded in the existing literature, particularly as it pertains to the diverse manifestations of PRP, including those affecting the scalp. The importance of understanding scalp involvement in PRP cannot be overstated, as it plays a critical role in both diagnosis and management. Trichoscopic examination, as highlighted in this report, provides valuable insights that can aid in distinguishing PRP from other scalp disorders, contributing to a more accurate and timely diagnosis. Additionally, the unique trichoscopic findings described here add to the growing body of knowledge about PRP, offering clinicians new tools for assessment. By continuing to enhance our understanding of PRP and refining both diagnostic approaches and treatment strategies, we can aim to improve the overall management and outcomes for individuals living with this challenging condition. The development of evidence-based algorithms and treatment protocols will be crucial in guiding clinical decisions and optimizing patient care in the future.

## Figures and Tables

**Figure 1 medicina-60-01839-f001:**
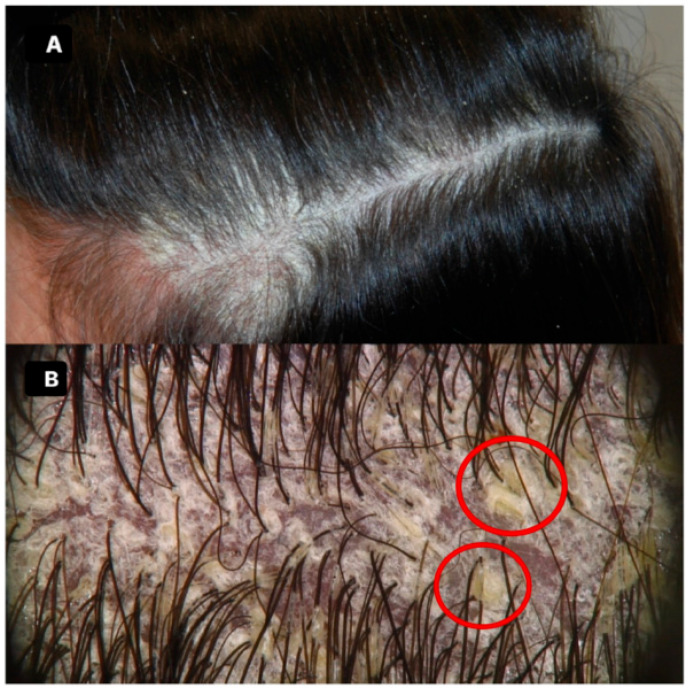
The global clinical picture (**A**) and trichoscopy (**B**) of our patient affected by scalp PRP: the clinical picture shows widespread scaling that organizes as a cap, while trichoscopy displays adherent white-silvery scales that form hair casts with a triangular shape, resembling an “arrowhead” that points towards the scalp (red circles); a few dilated and circular blood vessels, as well as some blood crusts, can be seen.

**Figure 2 medicina-60-01839-f002:**
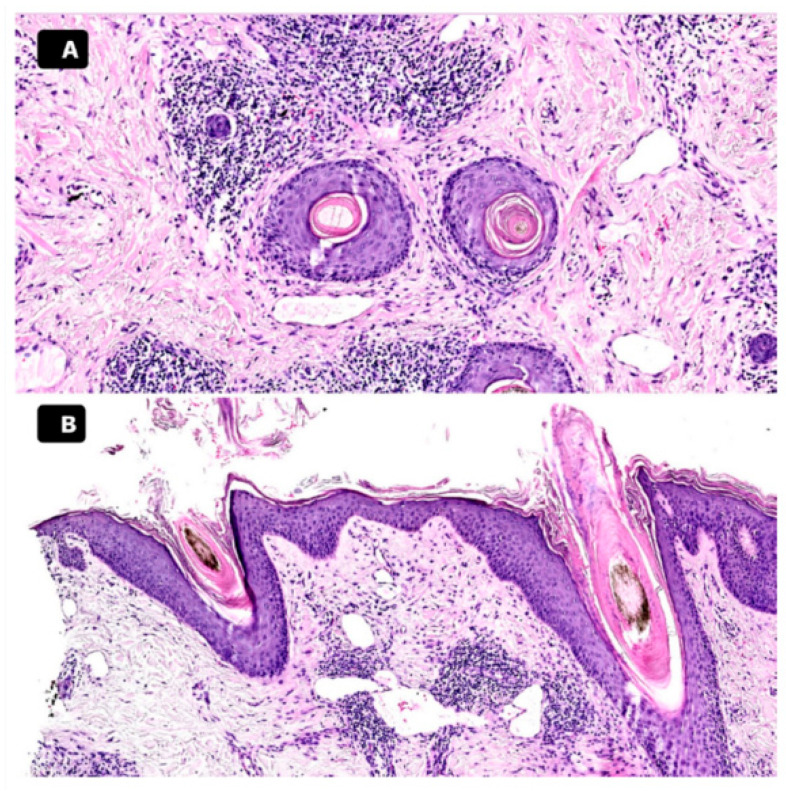
Histopathologic pictures. Dilated infundibula plugged by orthokeratotic and parakeratotic cells in alternation. A lymphocytic inflammatory infiltrate is present, primarily located around blood vessels and adnexal structures, along with multiple dilated vessels. H&E 10x. Horizontal section (**A**). Superficial perivascular dermatitis, psoriasiform. Dilated infundibula plugged by orthokeratotic and parakeratotic cells in alternation. H&E 10×. Vertical section (**B**).

**Table 1 medicina-60-01839-t001:** Summarized studies.

First Author, Year	Study Design	Number of Patients	Clinical Manifestations	Trichoscopic Features	Histopathological Features
Golińska et al., 2022 [26]	Case series	3	Scalp involvement in the erythrodermic variant of PRP	Dotted vessels, white and yellow scales and peripilar casts	NA
Sławińska et al., 2021 [21]	Case series	3	Scalp involvement in the erythrodermic variant of PRP	Dotted and small branched vessels over pinkish background, yellowish keratotic plugs	NA
Martín Callizo et al., 2014 [23]	Case report	1	Scarring alopecia		Significant perifollicular stellate fibrosis replacing the follicles, with no signs of inflammatory infiltrate. Direct immunofluorescence was negative.
Fernandez-Flores et al., 2021 [25]	Case report	1	Scarring alopecia	Erythematous patches with perifollicular casts and loss of follicular ostia	Scarring alopecia, with hair miniaturization and an increase in the telocatagen/anagen ratio, that involved the superficial reticular dermis. The fibrosis was distributed in a band parallel to the epidermis, better demonstrated in the examination with orcein staining for elastic fibers. The elastic network of the papillary dermis was preserved. Accentuation of the fibrosis in a peri-infundibular pattern, mild perivascular lympho-histiocytic infiltrate with discrete interface dermatitis involving the infundibular epithelium. No erosion of the basal layer of the interfollicular epidermis, and no dermal deposition of mucin. The superjacent epidermis was mildly acanthotic, with preservation of the granular layer and alternating parakeratosis in vertical and horizontal strata. No spongiosis. No fungi were found with periodic acid–Schiff (PAS) staining.
Lerebours-Nadal et al., 2016 [22]	Case report	1	Hyperkeratotic manifestation of scalp PRP in an HIV patient		Psoriasiform acanthosis and hyperkeratosis, with alternating horizontal and vertical ortho- keratosis and parakeratosis, as well as follicular plugging. In the dermis, mild lymphocytic inflammatory cell infiltrate.
Manoharan et al., 2006 [24]	Case report	1	Scalp scaling	NA	NA
Starace et al., 2024 [our case]	Case report	1	Diffuse white scaling	Closely adherent white-silvery scales forming hair casts with a triangular shape, resembling an “arrowhead” pointing towards the scalp; a few dotted, dilated and circular blood vessels.	Dilated infundibula plugged by orthokeratotic and parakeratotic cells alternately, along with superficial perivascular dermatitis.

**Table 2 medicina-60-01839-t002:** Differential diagnosis for PRP.

Disease	Clinical Manifestations	Trichoscopic Features	Histopathological Features	First Author, Year
Scalp psoriasis (SPso)	Thick, adherent and white-silvery scales regularly distributed across the scalp	White-silvery scales Erythema Dotted vessels (red dots) Globular and looped vessels Loss of follicular openings	Mounds of parakeratosis with neutrophils Spongiform micropustules of Kogoj Clubbed and evenly elongated rete ridges A decrease in the size and number of sebaceous glands An increase in catagen and telogen hairs Follicular miniaturization Perifollicular lymphohistiocytic inflammation Infundibular dilatation with thinning of the follicular infundibulum Tortuous blood vessels	Bruni et al., 2021 [32] George et al., 2015 [33] Park et al., 2016 [34]
Seborrheic dermatitis (SD)	Fine, greasy, loosely adherent and yellowish scales	Yellowish scales Follicular plugs Yellowish crusts Dotted, linear and arborized vessels	Follicular plugging Shoulder parakeratosis Prominent lymphocytic exocytosis Marked epidermal spongiosis predominantly at the follicular ostia Irregular acanthosis Abundant plasma in the mounds of parakeratosis Mounds of parakeratosis with neutrophils	1. Park et al., 2016 [34] 2. Dall’Oglio et al., 2022 [35]
Pseudotinea amiantacea (PA)	Tightly adherent, thin, silver-white scales that encase the hair shafts, forming a thick covering	Tightly adherent scales Not significant vascular changes	Diffuse hyperkeratosis and parakeratosis together with follicular keratosis, which surround each hair with a sheath of horn	1. Verardino et al., 2012 [36] 2. Abdel-Hamid et all., 2003 [37]

## Data Availability

The data that support the findings of this study are available on request from the corresponding author, S.C. The data are not publicly available due to their containing information that could compromise the privacy of the research participant.

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
