# Peer review of "Uncommon Presentation of Pityriasis Rubra Pilaris of the Scalp: Clinical, Trichoscopic, and Histopathologic Features and Review of the Literature"

_medicina, 2024, doi:10.3390/medicina60111839_

Round 1

Reviewer 1 Report

Comments and Suggestions for Authors

The comments for Case based Review  Uncommon Presentation of Pityriasis Rubra Pilaris of The  Scalp: Clinical, Trichoscopic and Histopathologic Features and Review of The Literature are as follows:

1.      Abstract need to elaborate with need of this review and what content it covers.

2.      Introduction need to improve with addition of latest work. There are two bulk repeated citation, 1-4 and 6-11. Please cite them at appropriate place, sentences separately.

3.      The data is limited and discussion need to improve with proper flow.

4.      By carefully analyzing these trichoscopic features, we were able to distinguish our 222 case of PRP from these differential diagnoses, each of which presents with a unique set of 223 signs that can be identified through trichoscopy. This work need to cited.

5.      There are lot many small paragraphs are there which not interconnected, and linked. Improve that.

6.      Conclusion need to improve with removing generalized statement like, PRP is a rare and complex inflammatory skin disease with a wide 285 range of clinical presentations and an unpredictable prognosis.

7.      If more illustrative data available please include.

8.      Institutional Review board Statement: Ethical review and approval were waived for this study in accordance with national regulations, which do not mandate such review for scientific articles. This need review again. 

Reviewer 2 Report

Comments and Suggestions for Authors

The article presents an intriguing case report on a 59-year-old female with Pityriasis Rubra Pilaris (PRP), discussing the clinical aspects as well as trichoscopic and histopathological findings. It compares these findings to existing literature, including differential diagnoses and treatment options. While the article holds significant potential and clinical interest, several adjustments could enhance its appeal to readers.

- I suggest the authors highlight the "arrowhead" lesion revealed by trichoscopy in Figure 1.a.

- Genes should be described in uppercase and italicized format, for example, CARD14. This should be corrected throughout the text.

- I recommend that the authors add a paragraph in the introduction clearly outlining how clinical diagnosis is typically performed (e.g., is there a grading of lesions?) and the differential diagnosis for PRP (how is the diagnosis concluded? Are there any molecular markers?), relating this to the importance of employing the techniques utilized in the study.

- I propose that the authors create a table differentiating clinical, trichoscopic, and histopathological findings of conditions that may be easily confused with PRP, such as SPso, SD, PA, etc. This would assist clinicians in identifying findings and making more targeted diagnostic and therapeutic decisions.

- In the introduction, I suggest the authors include a paragraph providing context about autoinflammatory diseases and their distinction from autoimmune diseases, considering that PRP is recognized as a variant of psoriasis, which is thought to be an immune system disorder with significant influences from determining genetic factors. In this matter, I also suggest the authors to cite the main molecular pathways involved in autoinflammatory and autoimmune diseases.

- Another recommendation is to discuss the predominant cell type in these lesions and the cytokines that activate these cells, thereby creating an immunological framework for the reader. After all, PRP is an inflammatory disease.

- The article discusses molecular aspects very minimally. A discussion on the genetic and molecular pathways involved in the pathogenesis of the disease would greatly enrich the paper, particularly if related to infections and cancer. For instance, could infections and alterations in proto-oncogenes/tumor suppressor genes be triggers or consequences of the chronic inflammation caused by PRP? In this context, not all mechanisms may be elucidated when considering PRP; however, many mechanisms in clinically similar conditions have already been described. Thus, the authors could propose that molecular pathways involved in these other described conditions may also be active in PRP.

Reviewer 3 Report

Comments and Suggestions for Authors

This is an interesting case report and some review points are mentioned, the presentation is unique, but little is discussed as to its significance. Please correct.

I have some comments:

Are the prevalences for paraneoplastic PRP alike for the 2 age groups (both for children under 10 years and adults in 5-6th decade)? If yes, should all children with PRP be evaluated for cancer?

In your case report you do not mention any workup to eliminate paraneoplastic PRP, no mention of therapeutic response or any progression, or disease history/outcome. It reads more like a description of one sole clinical symptom, the skin changes of the scale, rather than a case report.

As a teaching moment for your case report you only mention this new clinical presentation, claiming it is a new entity, but since this is only one case the conclusion that this is a new feature of PRP should be suggested, not claimed. Do you have an explanation for the specific appearance?

For the paper to be more useful to clinicians I also suggest a brief description of workup (for your patient, and for workup in general). In the Introduction, you mention different types, paraneoplastic, associated with HIV, but then come with no recommendations as to how to approach a patient with PRP.

Could you consider using arrows to describe the specific findings in the clinical and histological photos? In particular, could you point to the structures described as arrowheads and the dilated blood vessels?

The clinical photos are of good quality.

Reviewer 4 Report

Comments and Suggestions for Authors

The text describes an atypical case of scalp PRP (pityriasis rubra pilaris) and highlights its unique features compared to the literature. The authors presented a comprehensive diagnostic approach, combining clinical examination, trichoscopy, and histopathological analysis. This allowed for a thorough analysis of the disease. The identification of an unusual trichoscopic pattern (triangular scales resembling an arrowhead) is particularly important for future diagnostics, representing a valuable contribution to the understanding of PRP.The presentation of this unique case, which adds new insights to PRP diagnosis, is commendable. The authors should also be praised for their clear comparison of this case with the literature and their comparative analysis with other scalp conditions.
One suggestion for improvement would be to consider modifying the section on scalp psoriasis (SPso) in the discussion to make it slightly shorter and easier to undrestand. Additionally, the references are not formatted according to a consistent scheme.

Round 2

Reviewer 1 Report

Comments and Suggestions for Authors

All comments resolved. 

Author Response

Thank you!